# The CORA 5.2 dataset: global *in-situ* Temperature and Salinity measurements dataset. Data description and validation.

Tanguy Szekely<sup>1</sup>, Jérôme Gourrion<sup>1</sup>, Sylvie Pouliquen<sup>2</sup>, Gilles Reverdin<sup>3</sup>

<sup>1</sup>Societe Coopérative OceanScope, 115 rue Claude Chape, 29290, Plouzané, Brest

<sup>2</sup>IFREMER, BP 70, Plouzané, 29280, France

<sup>3</sup>Sorbonne -Université, CNRS/IRD/MNHN (LOCEAN), Paris, France. ORCID ID <u>https://orcid.org/0000-0002-5583-8236</u>

Correspondance to: Tanguy Szekely (tanguy.szekely@ocean-scope.com)

**Abstract:** We present the Copernicus *in-situ* ocean dataset of temperature and salinity (version V5.2). The ocean subsurface sampling varied widely from 1950 to 2017, as a result of changes in the instrument technology and development of *in-situ* observational networks (in particular, tropical moorings, Argo program). Thus the global ocean temperature data coverage on an annual basis grows from 10% in 1950 (30% for the North Atlantic basin) to 25% in 2000 (60% for the North Atlantic basin) and reaches a plateau exceeding 80% (95% for the North Atlantic Ocean) after the deployment of the Argo program. The average depth reached by the profiles also increased from 1950 to 2017. The validation framework is presented, and an objective analysis-based method is developed to assess the quality of the dataset validation process. Analyses of the ocean variability are calculated without taking into account the data quality flags (raw dataset OA), with the near real-time quality flags (NRT dataset OA) and with the delayed time mode quality flags (CORA dataset OA). The comparison of the objective analysis variability shows that the near real-time dataset managed to detect and to flag most of the large measurement errors, reducing the analysis error bar compared to the raw dataset error bar. It also shows that the ocean variability of the delayed time mode validated dataset is almost exempt from the random error induced variability.

Keywords: Global dataset, In-situ, Temperature and salinity profiles

#### 1. Introduction

Estimating the temperature and salinity ocean state is critical for documenting the evolution of the ocean and its role in the present climate. To do so, the scientific community relies on *in-situ* measurements at a global scale and integrable detected.

into global datasets.

Among the global datasets, one can cite the world ocean database (Boyer et al, 2013, hereafter WOD) and the EN4 database (Good et al. 2013, <u>www.metoffice.org</u>) distributed by the UK Meteorological Office. Here, we present CORA (Coriolis Ocean dataset for ReAnalysis), a dataset distributed by Copernicus Marine Service (hereafter CMEMS) and produced by Coriolis. CORA differs from these earlier datasets by choices in the

- construction and the production of the dataset. Indeed, WOD is validated with the highest quality control methods at 102 vertical levels, whereas the EN4 profiles are limited to a maximum of 400 vertical levels and is automatically validated (Ingleby and Huddleston, 2007). CORA conversely retains data at the highest vertical resolution. The choice of reducing the number of levels in the data validation and in the dataset, construction helps to quickly cluster new measurements to the dataset and provides easy to handle datasets. On the other hand, these
- methodologies result in a loss of measurements potentially available for the scientific community, through the vertical sampling of the profiles or in the data validation. In the construction of CORA, all the measurements available are kept, then an automatic validation is first performed followed by a manual/individual check (Gaillard et al. 2009, Cabanes et al, 2013). This validation framework requires the production of two datasets, a near real-time validated dataset, distributing the profiles within days after collection, and a delayed-time validated dataset,
- covering in year *n* the historical period up to year *n*-1. This choice, made in the early versions of CORA, has been retained in the latest one that we describe here.

The global ocean heat content (GOHC) increase has been observed on decadal time scales, whether it is in the upper layers of the ocean (Domingues et al, 2008, Ishii and Kimoto, 2009, Levitus et al, 2009), below the thermocline (Von Schuckmann and Le Traon, 2011) or in the abyss (Purkey and Johnson, 2010). Beside the

- influence of the mapping method and the baseline climatology (Abraham et al, 2013, Cheng and Zhu, 2015, Boyer et al. 2016, Gouretski, 2018), the data validation performed on *in-situ* measurements has a direct influence on the estimation of global ocean indicators such as GOHC, global freshwater content and sea level height (Abraham et al, 2013, Gouretski, 2018). As an example, differences in the GOHC estimation in the Johnson et al, 2010 analysis compared to the Lyman et al. (2010) analysis have been shown to result from quality control issues. The particular
- case of XBT measurements (Levitus et al, 2009, Cheng et al, 2009) influence on the GOHC estimation is well documented. Systematic errors in other instrument types may also introduce systematic biases leading to biases in the GOHC estimation (Lyman et al, 2006, Willis et al, 2011). The validation of a quality control method is thus a critical task to ensure that the dataset flags are accurate enough to flag erroneous measurements without biasing the dataset. The uncertainty surrounding the quality assessment of large oceanographic dataset being a
- critical topic in the ocean climate studies, we propose here a method of global dataset quality assessment and we apply it to the near real time validated and delayed time mode validated datasets. We will first list the data sources of the CORA measurements in section 2. A description of the CORA data space and time repartition will be reported on section 3. Then, the quality control procedure will be described in section

4. Lastly, gridded temperature and salinity fields are calculated using an objective mapping that is presented insection 5. The results of the dataset validation and quality assessment are finally discussed on section 6.

#### 2. Data providers

The CORA 5.2 dataset is an incremental version of the previous CORA datasets, covering the period 1950 to now
and distributed by CMEMS. Most of the CORA profiles are first collected by the Coriolis data center and validated
in near real time mode. Coriolis is a Global Data Assembly Centre (DAC) for the Argo program (Roemmich et al. 2009). It collects Argo profiles from the regional Data Assembly Centers (DACs) and distributes them to the community. Coriolis also collects XBTs, CTDs and XCTDs measurements from French and Europeans research programs as well as from the Global Telecommunication System (GTS), Voluntary Ship System (VOS),

- subtropical mooring networks (TAO/TRITON/RAMA/PIRATA programs from PMEL). A major effort has also been made to include smaller datasets to the Coriolis dataset that are available in delayed time mode, such as the ITP and CTD profiles from the ICES program, Sea Mammals measurements from MEOP (<u>http://www.meop.net</u>) and validated surface drifter data. Delayed time mode measurements have also been loaded from the Word Ocean Database (WOD13) and the French Service Hydrographique de la Marine (SHOM). It should be noted that in the
- case of a profile distributed by Coriolis in real time mode and by one of these datasets in delayed time mode, the delayed time mode validated profile replaces the real time mode profile in the CORA database.
  Last, recent comparison of the CORA profile positions with the EN4 dataset (metoffice.gov.uk) have shown that some of the profiles distributed in EN4 were not in CORA previous versions. A partnership with the EN4 teams allowed us to detect and to import most of those profiles. 5069864 profiles have been imported in this way,
- covering the period 1950-2015. However, contrary to the other measurements, the profiles from the EN4 database are not reported with a pressure measurement, but instead with depth and with a maximum number of reported levels in an individual profile set to 400. The issue of the inhomogeneity in the dataset with respect to the vertical sampling, will be discussed.

## 65 **3. Dataset description**

The CORA dataset aims to provide a comprehensive dataset of *in-situ* temperature and salinity measurements from 1950 to 2017. The oceanic temperature and salinity measuring instruments have however radically changed during the last 70 years. As a result, the origin and characteristics of data distributed in CORA dataset widely varied in time (Fig : 1) Most of the profiles collected prior to 1965 are mechanical bathythermographs (MBT) measurements or Nansen casts. From the late 1960s to 1990, the most common profile are from the expendable bathythermographs (XBT), developed during the 1960s and widely used by navies. Most of the XBT profiles

The development of the Sippican T-7 instrument with a maximum depth of 1000m slowly increases the number
of measurements between 460m and 1000m during the 1980s (see Fig: 2 for the dataset measurements distribution with depth). An instrument capable of measuring conductivity, temperature and pressure (CTD) was

collected during this period are T4 type sensor, measuring temperature above 460 meter depth.

developed in the 1960s, allowing an accurate estimation of sea salinity and temperature. The yearly amount of CTD profiles in the CORA dataset then slightly increased reaching a plateau of about 20000 profiles in the early 1990s.

- During this period, the largest density of profiles is found in the North Atlantic Ocean, with a coverage ratio, calculated on a 3° per 3° grid with a one year time step, increasing from 30% in 1950 to a plateau of 60-70% in the 1970s (Fig: 3). The North Pacific mean sampling rate is lower than 10% before 1965, with the largest portion of the collected profiles located close to the Japanese and North American coasts and along a transect connecting the USA West coast to the Hawaian archipelago (not presented). It quickly increases from 1965 to 1970 to reach
- about 50% in the early 1980s with a more homogeneous spatial resolution. Before 1974 in the other ocean basins, most of the collected profiles are found in the coastal zone and along a few ship tracks. The coverage then slightly increases in the western part of the Indian Ocean and in the eastern part of the South Pacific Ocean, increasing the associated basin sampling rate from 10 % in 1965 to 20-25% in 1990. The Austral Ocean sampling rate remains however around 5% during the whole period.
- During the 1990 decade, the yearly number of XBT profiles strongly decreases while the number of bottles and CTD profiles slightly increases. The counter-intuitive behavior is mostly caused by a lack of XBTs in the Coriolis database during the 1990s. The yearly number of XBTs should indeed decrease slowly during the 1990s and reach the CORA level by the end of the decade. This problem should however be fixed in the next version of CORA. The measurements provided are however deeper than in the previous decade, leading to a better coverage below
- 500m depth (Figure 2). The profile number then exponentially increases since the development of the TAO/RAMA/PIRATA equatorial mooring program throughout the 1990s. During this time, the North Atlantic and the North Pacific Ocean spatial sampling rates decreases, and the global ocean sampling rates reach a plateau at 20%. The ocean sampling rate rapidly increases in the early 2000s thanks to the development of autonomous profilers and the worldwide Argo program.
- The global ocean sampling rate reaches 70% before the mid-2000s with a maximum of 85% in the northern Atlantic Ocean. Notice the simultaneous growth of the autonomous profiler measurements (figure 1), and the increasing amount of measurements below 1000m depth on figure 2. In the Austral Ocean, the sampling is sharply increased from 8 to 40% in 2005-2006, and then grows slowly up to 50% in 2017. This increase in the Austral ocean coverage is a combined consequence of Argo deployments mostly north of 55° S and the collection from
- CTD casts mounted on sea mammals, in particular between Kerguelen Island and the Antarctic continent (Turpin et al. 2011).

It must be emphasized that a fraction of the profile numberincrease of the early 2000s results from the data acquisition from high frequency measurement devices such as the ocean drifters, the thermosalinographs (TSGs), both near the ocean surface, or undulating CTDs either towed or untowed (scanfish, seasoar, gliders,...). Indeed,

- each undulating CTD profile and each independent TSG or drifter measurement is treated as an independent profilewhile one could also cluster them by instruments of by cruise. The dataset structure we retained is however easier to handle by the ocean reanalysis community and leads to a more homogeneous dataset file structure. This dataset structure is also adopted for the mooring measurements which in some cases are also collecting data at high frequency. This large number of mooring data induces a large increase of measurements such as at 250m and
- 500m depths, whereas at the surface, the large increase is due to data from TSGs and drifting buoys.

## 4. Data quality control

The measurements collected by the Coriolis data center are distributed to the scientific community with a near real time quality control flag within days from the data reception and with a delayed time mode validation quality control within a year. The Coriolis datacenter validation workflow scheme is given on figure 4.

The quality control flags applied on the CORA dataset are associated to a measured or a calculated variable (TEMP\_QC, PSAL\_QC, DEPTH\_QC, PRES\_QC) and on the date and position variable (POSITION\_QC and JULD\_QC) and the associated adjusted variable when they exist. The QC flag values applied during the quality

- control process vary from 1 to 4, with 1: good data, 2: probably good data, 3: probably bad data and 4: bad data. Numerous measurements distributed by Coriolis have however been validated by scientific teams prior to the integration in the Coriolis streamflow. The most important of these datasets are the delayed time mode validated Argo profiles, the tropical mooring dataset, distributed by PMEL, the sea mammal measurements validated by the MEOP project and the TSG measurements validated by the GO-SUD project. In such cases, the current practice
- at Coriolis is to retain the flags from the imported database and to run the delayed time mode tests afterwards.

# 4.1 Near real time validation

The near real-time dataset validation tests are mostly taken from the Argo real time quality control tests (Wong et al. 2009). The goal is to distinguish the spurious measurements from the good measurements and to flag them quickly. The test checks are designed to detect well known types of errors. A global range test and a regional

- range test are performed to detect obvious errors with respect to known ocean variability. The bounds of those two tests are very large with respect to the known ocean variability to ensure that no bad flag would be incorrectly attributed. A spike test and a gradient test are performed to detect measurement spikes in the temperature and salinity fields. The test is based on the comparison of the temperature and salinity vertical gradient to a threshold. The test thresholds are set large enough to lower the number of incorrect spike detections corresponding to a
- sharp, yet correct, thermocline or halocline. The stuck value test aims to detect temperature or salinity profiles with a constant value within the vertical reported inaccurately.

A second step in the near real time quality control is performed daily on the Argo profilers distributed by Coriolis using an objective mapping detection method (Gaillard et al. 2009). Following the framework developed by Bretherton et al. (1976), the residual of the objective analysis depends on the covariance from data point to data

- point. Thus, this second check step aims at detecting measurements departing from other data in its vicinity. The correlation scale in the objective analysis varies with depth and latitude. Spurious detections can however occur when profiles located on both sides of a frontal zone are within a correlation radius. Therefore, detected profiles are visually checked by a PI to distinguish erroneous measurements from correct measurements.
- Lastly, a quality control based on altimetry comparisons is also performed on a quarterly basis to improve the real
   time validated dataset (Guinehut et al. 2009). A PI investigation is also performed on profiles flagged as suspicious by comparison with altimetric sea level.

#### 4.2 Delayed time mode validation tests

The delayed time mode validation is performed on a yearly basis. This validation framework is based on tests more stringent than the near real-time validation process, which requires a systematic visual control by an oceanographer. The controlled profiles are those which have not been controlled in the previous version of CORA.

Therefore, most of the controlled profiles for a given version of CORA are the profiles measured during the previous year, but not controlled for the earlier version. The profiles for which the measurements have been updated or adjusted since the latest version are however controlled. Last, some datasets covering the historical period may have been incorporated in the Coriolis dataset, which are then controlled in delayed time mode in

# 160 CORA.

The delayed time mode validation process is schematized on Figure 4. The profiles to be validated are first checked by the CORA tests. The checks raise an alert flag on suspicious profiles, which are then visually checked. For CORA, the validation checks are applied until all the tests are successful. If a single check test fails, the profile is put aside for visual check and the following tests are not applied. The profiles undetected by the CORA tests, and

165 thus not visually controlled, are assessed by a second set of tests developed by CLS. The suspicious profiles are also visually controlled and manually flagged. Last, all the tested measurements are gathered in the CORA database with the updated flags.

A first quality check aims to complement the real time QC procedure with redundant tests with sharper threshold than NRT.

# 170 Data-file consistency test

This test checks the obviously out of range position (|Lat|>90 and |Lon|>180 and out of range immersion (PRES>12000 decibar and Depth>12000 m or PRES<-2.5 decibar and DEPTH<-2.5 m). These tests are redundant with the NRT checks and are designed to avoid any writing error in the CORA file. The few detections are visually checked.

# 175 Depth check, Stability Check, Vertical check

The depth check, stability check and vertical check have initially been developed by the UK Met-Office for the EN4 dataset validation. They have been added to the CORA validation framework after a collaborative comparison of the two dataset validation methods with the UK Met-Office team. This study has shown that most of the profiles flagged in EN4 and not in CORA were detected by these three tests and that applying a visual

- control to the profiles detected in this way results in more accurate flags. The tests have been described in Ingleby and Huddleston, 2007. The stability test detects density inversions for profiles where both temperature and salinity are available. The density inversions with 0 > d\rho >-0.03 kg.m3 are dismissed. Both temperature and salinity are visualized for profiles with larger density inversion. Experience has shown however that most of the density inversions detected in this way are caused by small spikes in the salinity measurements, probably a consequence
- of anomalies in the conductivity measurement or alignment with temperature when estimating salinity. The spike test is designed to detect the temperature and salinity spikes and steps. It runs with a threshold of temperature and salinity variability varying from 5°C in surface to 1.5° C below 600 meter depth for temperature and from 1 PSU at surface and 0.2 PSU below 300 meter depth for salinity. These tests differ from the real time QC test since the trigger points are lower. They however sometimes create `false positive' detection either by detecting the wrong
- point on a spurious profile or by detecting a correct measurement. A systematic PI visual flag selection is then performed on each of the detected profiles. *Level disorder and duplicated levels*

The profiles with a non-monotonous PRES or DEPTH vector are detected and the PRES or DEPTH vector are flagged in order to be monotonous. This test has been requested by the CORA end users, the oceanographic reanalysis community, to have a user friendly dataset to work with. Most of the detected profiles are indeed

- measurements with a very slow sinking speed near the surface, giving pressure vector inversion when exposed to the sea surface swell. Most of the detections are thus confined to the surface layer. Exceptions may however occur in the case of Black Sea Argo floats for which a recurrent problem of slow sinking speed is found at sub surface due to the low salinity level of the Black Sea. Last, "hedgehog" type profiles, with very spiky temperature, salinity and pressure vectors, which are often caused by transmission mistakes on Argo floats, are detected by this test.
- Global range

The global range test aims to detect obvious measurement mistakes. The Temperature measurements under -3 °C or over 43°C and the salinity measurements under 0 PSU or over 46 PSU are detected. This test has a very low detection rate, but it still detects some erroneous profiles each year. Most of them are profiles with a non-classical shape so that they avoid detection by redundant tests (Minmax test or climatological test). A recent example was

205 an Argo float grounded near Mogadicio, Somalia, measuring a temperature exceeding 43°C, whereas the corresponding pressure was just above 0 decibar, so that the measurement avoided the other NRT and delayed time mode tests confined to depths between 0 and 2000 m.

The following step of the CORA data validation is performed in the Coriolis datacenter to detect profiles diverging from the known ocean variability. Each temperature and salinity profile is compared with the minimum and

- maximum measured value reference profiles. Those profiles originate from reference fields on a gridded mesh with 1 degree resolution horizontal hexagonal cells of 20m thickness. The reference fields are the maximum and minimum measured values on a set of 1.2 million Argo profiles, vertically interpolated from the surface to 2000 m depth. The field coverage is increased, especially in the inner seas and in the Austral Ocean, badly covered by the Argo network, by CTDs from the World Ocean Database and sea mammals measurements from the MEOP
- database The CORA 5.2 measurements are compared to the minimum and maximum reference values of the corresponding cell and the upper and lower adjacent cells in the same grid column. The profiles containing measurements exceeding the reference values are checked by an oceanographer. The minmax method is relaxed on the continental shelf since the minmax sampling is insufficient in the continental shelf zones. The temperature and salinity profiles measured over a bathymetry inferior to 1800m are compared to a climatology field (ARIVO,
- Gaillard et al) to which is added or subtracted 10 times the climatological standard deviation field. This criterion have been added since the minimum and maximum reference field is mostly based on the ARGO measurements and is ill defined in the ocean regions the Argo floats struggle to reach. Due to the lack of accuracy of the global climatologies near the coasts and on some shelves, the profiles lying above the 250 m isobath are not tested with this method. See Gourrion et al. 2019 for further discussion on the minmax field.
- A third validation is performed with the ISAS objective analysis tool, following the method developed by Gaillard et al, (2009). During the objective analysis process, the profile analysis residual is compared to the analysis residual of neighboring profiles. This validation test is similar to the validation performed by the Coriolis datacenter in near real time with the Argo floats. The scope of the test is however extended to other profiles (XBT, CTD, etc...) and to Argo profiles which have been updated in the Coriolis database too late to be part of the near 230
- real time validation.

A last set of delayed mode validation tests has been developed by the CLS research and development team and aims to complement the validation tests. These tests provide a sharper expertise on bias detection, spike detection and ocean variability in the continental shelf zones. These tests also aim to complement the Coriolis real time

quality check tests for measurements directly included in the delayed mode dataset. The CLS tests are divided in

- two categories. A density check test is applied to detect small density inversions in the measurement. This test differs from the Coriolis density inversion test since it focuses on single point spikes on density profiles instead of checking spikes or steps on temperature and salinity profiles, with a simple yet reliable algorithm. This test is reliable so the detected suspicious levels are automatically flagged. A second set of tests is applied to detect smaller errors. These tests aim to detect unlikely extremes in temperature and salinity by comparing measurements
- to regional upper and lower bounds and World Ocean Atlas 2013 climatology. Tests are also applied on vertical density inversions, spikes and offsets with respect to the climatology. By the end of the validation process, about 10% of the applied flags are based on CLS detection and 90% are based on Coriolis detections.

# 5. CORA 5.2 quality control results

#### 245

The relevance of ocean climate studies strongly depends on the accuracy of ocean measurements. Systematic data errors might thus result in biasing the estimation of ocean state indicators such as the GOHC, the global ocean freshwater content or the global mean steric height (Levitus et al. 2009). Furthermore, random measurement and data error may lead to overestimate the ocean variability. Therefore, indirectly, one can assess the reliability of the global dataset by estimating the influence of the quality control on global metrics such as the ocean mean temperature and salinity and the associated variability.

Two mappings of ocean temperature and salinity based on the CORA dataset measurements are calculated: a raw estimation (GOHCraw) which considers every measurement without taking the data quality flags and a flagged estimation (GOHCflg) which only consider the good and probably good QCs.

Interpolated fields are calculated following the method presented by Forget and Wunch, 2007 that has the advantage of not biasing mean fields and not relying on specifying them. The global ocean is divided in 1° per 1° grid cells with 10 m vertical layers from the surface to 1500 m depth. A first estimation of the mean parameter for
a given month is given by calculating the mean of the temperature or the salinity data measured in a given cell. The variance field is estimated by taking the variance of the measurements located in a given cell, if the number of available measurements is greater than 4.

A spatial weighting function is defined:

$$G(i,j) = e^{\frac{(l_p - l(i,j))^2}{r_l^2} \frac{(L_p - L(i,j))^2}{r_L^2}}$$
[1]

With  $r_l$  and  $r_L$  latitude and longitude decorrelation scales, both taken equal to 5° at any point of the ocean, and  $l_p$  and  $L_P$  the latitude and longitude of a grid point.

The combined mean is then:

 $\overline{T}(i,j) = \sum_{p} \frac{G_{p}(i,j) \overline{T_{P}} n_{P}}{N(i,j)}$ [2]

With:

$N(i,j) = \sum_{P} G_{P}(i,j)n_{P}$ [3]

The combined variance is estimated with a similar operator.

$$var(T)(i,j) = \sum_{p} \frac{G_{p}(i,j) var(T_{P}) n_{P}}{N(i,j)}$$
[4]

With *nP*the number of measurements available in the summed grid point, *Tp*- the mean temperature at the grid point and N(i,j) the total number of measurements involved in the calculation of a grid point value.

The values of rl and rL are set to 5° longitude and latitude in order to include enough grid points with data in this averaging. To reduce the calculation time of the analysis, each N(i,j) calculation is performed on a 20 per 20 grid point window.

The objective analysis is performed at three steps of the global dataset. A first analysis is performed on a raw dataset, considering all available profile measurements. All the QC flags are considered good. A second analysis is performed on the same data profiles considering the QC available on NRT mode. A third one is performed on the same profiles considering the QC available on delayed time mode.

The ocean data coverage is sometimes insufficient to perform the monthly objective analysis on the whole ocean. As a result, we have limited this study to the latitude between  $60^{\circ}$ N and  $60^{\circ}$ S since the ocean data coverage is too sparse out of these limits, leading to random anomalies in the temperature and salinity variability. Figure 5 shows 295 an estimation of the ocean layer covered by the objective analysis as a percentage of the ocean layer surface between 60°N and 60°S. It shows that the ocean coverage is higher for temperature than for salinity objective analysis. The upper layers coverage are very close. It varies from 95% in 2005 to over 98% after 2012. The 1475-1525m depth layer departs from the others since it has a global coverage lower from the others, starting from 65%

in January 2005. It converges to over 98% after 2014. A monthly variability is observed in the Argo development period (2005-2010). It is probably caused by the slow arrival of Argo profilers in the southern zones. This behavior lasts up to 2012 in the deeper layer.

Fig: 6 shows the percentage of good and probably good QC flagsin the NRT and CORA datasets compared to the RAW dataset. It shows that the proportion of good and probably good flags yearly tendencies are almost the same 305 at all depth. Moreover, in any case, the CORA and NRT differs by less than 0.5%. The proportion of good and probably good temperature flags varies from a minimum of 92% in 2006 to a plateau of about 98% after 2013. The 975-1025 m depth and 1475-1525 m depth layers depart from the others with 1 to 2 % lower rate between 2005 and 2013. Punctual decrease of good and probably good temperature flag rates are observed in late 2007, late 2012, late 2014 and in the beginning of 2016 for the surface and subsurface layers. These spikes are caused 310 by a sharp increase in the number of profiles distributed from a tropical mooring from the RAMA network. These profiles are indeed first distributed in the Coriolis dataset as TESAC profiles transmitted from the GTSPP. The profiles corresponding to tropical moorings are usually later replaced by the corresponding measurements transmitted by PMEL and the TESAC profiles are deleted from the database. In this particular case, the TESAC profiles had been retained and flagged as bad profiles instead. The yearly number of profiles in the RAW dataset 315 is thus strongly increased but the corresponding number for the NRT and CORA dataset is not. The good and probably good salinity flag rate tendency is opposite to the good temperature flag rate, with a maximum of over 98% before 2010, then a decrease to a level of about 94% with a high interannual variability after 2011.

The mean 0-50m, 75-125 m, 275-325 m, 475-525 m, 975-1025 m and 1475-1525 m depth salinity standard 320 deviations analyzed by the method (eq. 4) from 2005 to 2016 are shown on figures 7 and 8. The mean salinity standard deviation is averaged between  $60^{\circ}$ N and  $60^{\circ}$ S for each dataset analysis. The comparison of the raw dataset analysis with the NRT analysis and the CORA analysis shows the gain in dataset quality resulting from the QC performed. In the raw dataset analysis, numerous random mistakes result in a high average salinity standard deviation. The raw dataset standard deviation is however lower in the early period at almost all levels, 325 despite a rather high global level for the 475-525 m depth layer and a variability spike in late 2006 in the bottom layer. This lower variability level is probably a consequence of the Argo program development from 2005 to 2008, the low coverage in the southern oceans preventing the emergence of high level values. During this period, a large seasonal variability is present in the upper layers, varying from an order 0.2 PSU during winter to 0.4 PSU during summer, in the surface layer. The peaks in the ocean variability are thus correlated with peaks of ocean 330 coverage (see figure 5). The objective analyses also have a higher proportion of ship-borne measurements, CTDs for instance, essentially made during summer, compared to the autonomous measurements these years. We can thus assume that the lower number of profiles during winter does not allow to sample correctly the subsurface ocean fronts, leading to an underestimated winter time ocean variability. The increase in Argo float data from 2005 to 2008 slowly decreased this bias in the ocean variability estimation. The raw dataset surface salinity 335 standard deviation increases during the 2010-2016 period at all depth levels, with an order 0.6 PSU amplitude and

spikes up to 1.2 PSU in 2010 in the surface layer, and spikes varying from 0.9 to 1.2 PSU in the other layers.

The NRT analysis is very close to the CORA analysis before 2008. This behavior is a consequence of the low number of measurements corresponding to this period collected or updated in the database after the validation of

- the last version of the CORA dataset. The flags in the NRT and CORA datasets are indeed the same except if an updated version of a profile is loaded in the database of if a new profile is loaded in the Coriolis database. On the other hand, large discrepancies between the NRT and the CORA datasets are recorded between 2009 and early 2012 and between late 2013 and 2016. Another fraction of the discrepancy between the NRT and the CORA error bars are caused by non-Argo profiles updated in the Coriolis database without delayed time mode assessment.
- Most of these measurements are sea mammal profiles in the northern Pacific Ocean or mooring data, imported from the GTSPP (Global Temperature and Salinity Profile Program, <u>https://www.nodc.noaa.gov/GTSPP/</u>) TESAC messages with biased salinity sensors. Sea gliders with an order 5 to 10 PSU bias in salinity were also documented. Moreover, despite a lower number of profiles flagged, a few CORA flagged Argo profiles have biases large enough to strongly increase the analyzed ocean variability. Some of the spikes in the NRT ocean
- variability documented in the upper layers, the late 2007 to 2008 spike for instance, are observed in the surface layers since they are caused by biased instruments operating at surface to subsurface layers. Some other spikes in the ocean variability, the 2009-2011 spike for instance, are caused by biased Argo measurements and thus impact the ocean variability from the surface to 2000 m depth.
- A striking feature is the corresponding spike visible in the NRT analysis and in the raw dataset analysis in late 2010, which suggest that major data errors have not been flagged in the dataset during the NRT validation. Further exploration of this anomaly has shown that a fraction of the larger error bar in the NRT analysis is caused by an issue in the update of delayed time mode processed Argo profiles. In a few cases when salinity measurements present large drifts, the Argo PIs can decide that the salinity drift is too high to be adjusted. In these cases, the PI provides to the global DAC a delayed time version of the profiles with an adjusted temperature field, but with a practical salinity field filled with fillvalues and a salinity QC field filled with "4" values (bad measurement status).
- In some cases, the Coriolis data center had updated the profiles by getting the temperature adjusted field but without creating a salinity adjusted field. The available salinity field and QC in the Coriolis datacenter is therefore the original salinity field which might not have been flagged at "4". In this study, a handful of these profiles, often associated with large salinity measurement drifts (for instance salinity values on the order 20 PSU in the Indian Ocean) have produced large error bars in the NRT analysis fields. This issue will be soon tackled in the Coriolis database.

The CORA analysis salinity standard deviation slowly varies in time, with an order 0.15 PSU in the surface layer,
an order 0.1 PSU in the 75m depth – 125m depth layer and an order 0.08 PSU in the 275-325m depth layer and below 0.05 PSU in the deeper layers. This behavior is a consequence of the delayed time mode validation process which strongly reduces the number of random mistakes in the dataset. This variability is probably a function of the local data resolution, the oceanic variability and measurement errors. The slow variability of the CORA salinity standard deviation and its reasonable range suggests that remaining errors in the dataset will not have a large importance. Thus this product is likely to present a low error amplitude.

Figures 9 and 10 show time series of the mean temperature standard deviation of the CORA, NRT and RAW analysis. As anticipated, the mean temperature standard deviation time series is noisy and rather high in the RAW dataset case. The mean amplitude varies almost linearly between 1.2 °C in the 0-50 m depth layer and 0.4°C in the 1475-1525 m depth layer, except for the 975-1025 m depth layer with a 1.2 °C spike. A striking feature is the

- decreasing mean temperature standard deviation amplitude in time for the RAW analysis. The reason of this behavior is rather unclear. One shall assume that the overall quality of the oceanographic *in-situ* temperature measurement improves because of improvements in the temperature sensor. On the other hand, it might also be the decrease in the number of deployed XBTs in the 2010s that reduces the number of random errors in the dataset, since the XBT instruments are known to produce erroneous measurements when they are not handled properly.
- The NRT analysis and CORA analysis time series are rather close in all the analyzed layers, except for a 0.8 1°C spike in 2014-2015, detected in all layers but the 1475-1525-m depth layer, and enhanced in the 475- 525-m and in the 975-1025-m depth layers. This anomaly is related to the flag of numerous XBT measurements during the CORA delayed time mode validation process. XBTs are indeed more likely to fail (spikes or bias caused by a stretching of the XBT wire or a contact between the XBT wire and the ship hull), or bad estimation of the
- measurement depth.. Most of the flagged XBTs are moreover T-4 and Deep blue models. These models are usually not measuring *in-situ* temperature below 460 m depth and 760 m depth respectively, leading to correlated anomalies in the upper layers with no impact on the ocean variability below 800 m depth.

The CORA analysis variability has a mean amplitude of 0.85°C with a clear seasonal cycle of about 0.3°C in the 0-50-m layer. The CORA analysis mean variability amplitude averages 0.95 °C in the 75-125 m depth layer, with a monthly variability uncorrelated with the seasonal cycle. The seasonal cycle amplitude is null in the deeper layers, with a CORA analysis mean variability amplitude of 0.6°C in the 275-325 m depth layer, 0.4°C in the 475-525 m depth layer, 0.2°C in the 975-1025 m depth layer and 0.1°C in the 1475-1525 m depth layer. The higher frequency variability decreases with depth and is almost null in the deeper layers as seen on figure 9 and 10. The noisy shape of this high frequency variability is probably a result of ocean monthly variability and the changing locations of the ocean profiles.

The 2014-2015 spike in the ocean variability, detected in all the layers except for the deeper one in the NRT analysis, is caused by many XBT profiles. Most of those profiles are deployed in the Indian Ocean across a transect linking the Gulf of Aden to Perth, Australia, corresponding to measurements performed by the ship of opportunity program (Goni et al. 2009). The profiles have been extracted from the World Ocean Database and have thus not

- been validated with the Coriolis real-time validation framework. Many biases and spikes, probably due to issues with the probes or with poor insulation of the XBT wire problems, have been flagged in delayed time mode. The largest part of the upper layer spikes in the NRT and RAW analyses is a result of these erroneous measurements. In addition to the usual issues with the XBT measurements, the profiles sometimes indicated negative values at subsurface depth or temperature of 36.269°C at depth located above the maximum functioning depth of the XBT
- (460 m depth for T-4 and T-6, 760 m depth for Deep Blue). These unrealistic values have not been flagged after the extraction from the WOD dataset, resulting in exponential growth of the local amplitude of temperature standard deviation in the RAW and NRT analysis in the 475-525 m depth and 975-1025 m depth layers.

A closer look at the vertical profiles of the temperature and salinity mean variability (Figure 9 and 10) shows that the CORA analysis temperature and salinity variability is far smaller than the RAW analysis and the NRT analysis

- estimation. The depth variability of the temperature and salinity mean variability is moreover closer to the expected oceanic variability, with a maximum ocean variability at the surface or close at sub surface with decreasing variability below the ocean mixed layer depth. We however lack a reference high quality dataset to compare with to prove that the CORA dataset is not decreasing the global ocean variability by over-flagging good data. . Indeed, one should keep in mind that most of the flags applied on these profiles are manually applied by
- physical oceanographers after receiving a detection alert, and that the rate of flagged profile in the CORA analysis is lower than the rate announced for a reference dataset and analysis based on automatic quality control tests (Gouretski et al. 2018).

#### 6. Conclusion

The CORA dataset is an extensive dataset of temperature and salinity measurements. Efforts have been made to provide the scientific community withinformation as close as possible from the physical measurement and to perform a strict quality control on all profiles. The CORA dataset indeed stands out from the EN4 dataset since the delayed time mode validation is based on automatic detections and systematic PI decision, reducing the number of mistaken bad flags. In addition to that, the profiles are not subsampled and the time series (TSGs and bill and be an effective of the time series (TSGs and bill and be an effective of the time series (TSGs and bill and be an effective of the time series (TSGs and bill and be an effective of the time series (TSGs and bill and be an effective of the time series (TSGs and bill and be an effective of the time series (TSGs and bill and be an effective of the time series (TSGs and bill and be an effective of the time series (TSGs and bill and be an effective of the time series (TSGs and bill and be an effective of the time series (TSGs and bill and be an effective of the time series (TSGs and bill and be an effective of the time series (TSGs and bill and be an effective of the time series (TSGs and bill and be an effective of the time series (TSGs and bill and be an effective of the time series (TSGs and bill and be an effective of the time series (TSGs and bill and be an effective of the time series (TSGs and bill and be an effective of the time series (TSGs and bill and be an effective of the time series (TSGs and bill and be an effective of the time series (TSGs and bill and be an effective of the time series (TSGs and bill and be an effective of the time series (TSGs and bill and be an effective of the time series (TSGs and bill and be an effective of the time series (TSGs and bill and be an effective of the time series (TSGs and bill and be an effective of the time series (TSGs and bill and be an effective of the time series (TSGs and bill and be an effective of the time ser

- drifters) are distributed. It also stands out from the WOD dataset since all measurements within a profile are validated in delayed time mode, reducing the number of mistaken measurements.
  - Moreover, this study develops an innovative method to assess the overall quality of a dataset. This method shows the improvements of the dataset quality flags thanks to Coriolis real time QC and the CORA delayed time mode
- QC frameworks. This method however lacks a comparison with an analysis based on other datasets to ensure that the CORA validation framework is not constraining its description of the ocean variability by over flagging good measurements. This discussion shall be further pursued. This method is based on the mapping of the Ocean variability. It is thus implicit that the ocean sampling is homogeneous and sufficient to perform a monthly analysis. These conditions are met at a global scale and for the ocean measurements from surface to 2000 m depth since
- the full deployment of the Argo network. Last, the ocean data coverage is however insufficient to have a global coverage before 2005 (see Fig.3 for the ocean basin data coverage ratio), especially at depth larger than 1000 m between 1990 and 2005 and at depth larger than 500 m before 1990, as seen on Fig.2. The method will thus have to be adapted to the ocean data coverage to provide a synoptic view of the dataset quality.

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

#### 455

Bretherton, F. P., Davis, R., Fandry, C.: A technique for objective analysis and design of oceanographic experiments applied to MODE-73. Deep Sea Research and Oceanographic Abstracts, 23, 559-582, 1976.

Boyer, T.P., Antonov, J.I., Baranova, O.K., Coleman, C., Garcia, H.E., Grodsky, A., Johnson, D.R. Locarnini,
R.A., Mishonov, A.V., O'Brien, T.D., Paver, C.R., Reagan, J.R., Seidov, D., Smolyar, I.V. and Zweng, M.M.:
World Ocean Database 2013. Silver Spring, MD, NOAA Printing Office, 208pp. (NOAA Atlas NESDIS, 72),
2013.

Boyer, T.P., Domingues, C.M., Good, S., Johnson, G., Lyman, J., Ishii, M., Gouretski, V., Willis, J., Antonov, J., Wijffels, S., Church, J., Cowley, R., Bindoff, N.: Sensitivity of global upper-ocean heat content estimates to

mapping methods, XBT bias corrections and baseline climatologies, Journal of climate, 28, 4817-4842, 2016

- Cabanes, C., grouazel, A., Von Schuckmann, K., Hamon, M., Turpin, V., Coatanoan, C., Paris, F., Guinehut, S., Boon, C., Ferry, N., De Boyer Montegut, C., Carval, T., Reverdin, G., Pouliquen, S., le Traon, P.-Y.: The CORA dataset: validation and diagnostics of *in-situ* ocean temperature and salinity measurements. *Ocean Science*, 9,1-18, 2013.
- Cheng, L., Zhu, J.: Influence of the choice of climatology on the ocean heat content estimation, *Journal of atmospheric and oceanic technology*, 32, 388-394, 2015.
  - Cheng, L., Abraham, J. Goni, G., Boyer, T., Wijffels, S., Cowley, R., Gouretski, V., Reseghetti, F., Kizu, S., Dong, S., Bringas, F., Goes, M., Houpert, L., Sprintall, J., Zhu, J.: XBT Science: Assessment of instrumental biases and errors. American Meteorological Society, June 2016, 923-933, 2016.
- Domingues, C. M., Church, J. A., White, N. J., Glecker, P. J., Wijffels, S. E., Barker, P. M. and Dunn, J. R.: Improved estimates of upper ocean warming and multi decadal sea-level rise, Nature, 453, 1090-1093, doi:10.1038/nature07080, 2008.
  - Forget, G., Wunch, C.: Estimated global hydrographyc variability. *Journal of physical oceanography*, 37,1997-2008, 2006.
- Gaillard, F., Autret, E., Thierry, V. Galaup, P., Coatanoan, C. Loubrieu, T. : Quality control of large Argo datasets, Journal of Atmospheric and oceanic Technology, 26, 337-351, 2009.

Gaillard, F., Charraudeau, R.: New climatology and statistics over the global Ocean, MERSEA-WP05-CNRS-STR-001-1A, 2008.

Goni, G.: The Ship of Opportunity Program. The Information Society - TIS. 10.5270/OceanObs09.cwp.35, 2009.

Good, S. A., Martin, M. J. and Rayner, N. A.: EN4: Quality controlled ocean temperature and salinity profiles and monthly objective analyses with uncertainty estimates, *Journal of Geophysical Research: Oceans*, 118, 6704-6716, 2013. Gouretski, V.: World Ocean Circulation Experiment – Argo global hydrographic climatology, *Ocean Science*, 14, 1127-1146, 2018.

Gourrion, J., Szekely, T., Reverdin, G.:. The minmax field, to be submitted to JAOT, 2019.

- Guinehut, S., Coatanoan, C., Dhomps A.-L., Le Traon, P.-Y., Larnicol, G. : On the use of satellite altimeter data in Argo quality control. *Journal of Atmospheric and oceanic Technology*, *26*, *395-402*, *2009*.
  - Hill, C., Menemenlis, D., Ciotti, B., and Henze, C.: Investigating solution convergence in a global ocean model using a 2048-processor cluster of distributed shared memory machines. *Sci. Programm.*, Volume 15, Issue 2, Pages 107-115, 2007.
- Ingleby, B. and Huddleston, M.: Quality control of ocean temperature and salinity profiles- Historical and realtime data. *Journal of marine Systems*, 65, 158-175, 2007.
- Ishii, M. and Kimoto, M.: Reevaluation of historical ocean heat content variations with time-variing XBT and MBT depth bias corrections, J. Oceanogr., 65, 287-299, doi:1007/s10872-009-0027-7, 2009.
- Johnson, G. C., Lyman, J. M., Willis, J. K., Levitus, S., Boyer, T., Antonov, J., Good, S. A.: Global oceans: Ocean heat content, in State of the Climate in 2011, von 93, edited by J. Blunden, and D. S. Arndt, Bulletin of the American Meteorological Society, ppS62-S65, Boston, MA, doi:10:1175/2012BAMSStateoftheClimate, 2012.
- Levitus, S., Antonov, J., Boyer, T., Locarnini, R., Garcia, H., Mishonov, A.: Global ocean heat content 1955–
  2007 in light of recently revealed instrumentation problems, *Geophysical Research Letters*, 36, L07608, 2009.
  - Levitus S., Antonov J., Baranova, O., Boyer T., Coleman C., Garcia H., Grodsky A., Johnson D., Locarnini R., Mishonov A., Reagan J., Sazama C., Seidov D., Smolyar I., Yarosh E., Zweng M.: The World Ocean Database, Data Science Journal, 12, WDS229-WDS234, 2013.
  - Lyman, J. M., Willis, ? J. K., Johnson, G. C.: Recent cooling in the upper-ocean, *Geophysical Research Letters*, 33, L18604, doi:10.1029/2006GL027033, 2006.
    - Lyman, J. M., Good, S. A., Gouretski, V. V. Ishii, M, Johnson, G. C., Palmer, M. D., Smith, D. M. and Willis, J. K.: Robust warming of the global upper ocean, Nature, 465, 334-337, doi:10.1038/nature09043, 2010.
    - Purkey, S. G. and Johnson, G. C.: Warming of global abyssal and deep Southern Ocean waters between the 1990s and 2000s. J. Clim., 23, 6336-6351, doi:10.1175/2010JCLI-D-11-00612.1, 2010.
- Raper, S. C. B., Gregory, J. M. and Stouffer, R. J.: The role of climate sensitivity and ocean heat uptake on AOGCM transient temperature response, J. Clim., 15, 124-130, 2002.
  - Roemmich, D., Johnson, G., Riser, S., Davis, R. Gilson, J., Owens, W., Garzoli, S., Schmid, C., Ignaszewski, M.: The Argo program: Observing the global ocean with profiling floats. *Oceanography*, 22, 34-43, 2009.
- Roquet, F., Charrassin, J.-B., Marchand, S., Boehme, L., Fedak, M., Reverdin, G., Guinet, C.: Delayed-mode
   calibration of hydrographic data obtained from animal-borne satellite relay data loggers. *Journal of Atmospheric and oceanic Technology*, 28, 787-801, 2011.
  - Von Schuckmann, K. and Le-Traon, P.-Y.: How well can we derive global ocean indicators from Argo data? Ocean Sci., 7, 783-791, doi:10.5194/osd-8-999-2011, 2011.

Willis, J. K., Lyman, J. M., Johnson, G. C.: Correction to "Recent cooling of the upper ocean", *Geophysical Research Letters*, 34, L16601, doi:10.1029/2007GL030323, 2007.

Wong, A., Keeley, R., Carval, T.: Argo quality control manual, 2009.

Wong, A., Johnson, G., Owen, W.: Delayed mode calibration of autonomous CTD profiling floats by theta-S climatolog. *Journal of atmospheric and Oceanic technology*, 20, 308-318, 2002.

530

525

535

540

Figure 1: Yearly number of distributed profiles, sorted by instrument types.

Figure 2: Yearly number of measurements as a function of depth.

Figure 3: Yearly filling ratio of 3° latitude per 3° longitude gridded field of ocean basins