# Peer review of "The CORA 5.2 dataset: global *in-situ* Temperature and Salinity measurements dataset. Data description and validation."

_Ocean Science, 2018_

## Referee Comment (RC1) · Anonymous Referee #1 · 6 Feb 2019

Review of

The CORA 5.2 dataset: global in situ Temperature and Salinity measurement dataset. Data description and validation.

by T. Szekely et al.

[Figure]

**Recommendation**

**Minor revisions.**

**Synopsis**

CORA is the Copernicus *in-situ* data set of temperature (T) and salinity (S). Version 5.2 covers the period 1950-2107. The data consist of vertical profiles from XBT, CTD, XCTD, Argo, and moorings as collected and validated by the Coriolis data centre. As the title implies, the paper is a thorough description of the data sources and the applied quality controls (QC). Through a cooperation with EN4 new profile data have been incorporated into the data base.

Although most data have been quality-controlled by their PI before being added to the Coriolis database, all are passed trough the QC process again. The separate QC steps are described. The largest change comes from replacing Argo data that have undergone near-real time QC (automated) directly after becoming available by (semi-automatically) delayed-mode QC'ed values. The impact of the additional QC effort on the resulting data set is described. A notable result is a large reduction and homogenisation (in time) of the variability of the dataset, suggesting that no large errors remain in CORA 5.2.

**Discussion**

Users of a dataset or a collection of datasets should know how the data were collected and prepared, and which issues were tackled during the QC process. This information is given for CORA 5.2 in this paper, and it should therefore be published.

The main data source for the recent years is Argo. Therefore, the authors should add a short paragraph explaining the relation between CORA and the recently published Argo climatology (Gouretski, V.: World Ocean Circulation Experiment – Argo Global Hydrographic Climatology, Ocean Sci., 14, 1127-1146, https://doi.org/10.5194/os-14-1127-2018, 2018).

At the end of sect. 5 the authors state that they do not think that their quality-controlled data are overflagged. This is a very important conclusion, and it should receive more attention. Especially, the reasons given for this conclusion should be backed by more evidence.

**Detailed comments**

The paper would benefit from thorough language editing.

**p 2, l 5-7** three times "scientific community" - boring, please reformulate

**p 2, l 23** timeseries → time scales

**p 2, l 25** Baseline → baseline

**p 4, l 92** barely → slightly? I am not sure what you want to say.

**p 4, l 96** barely maintain a plateau at 20% → reach a plateau just below 20%? Again I am not sure what you want to say.

**p 5, l 125** for each of the test described in this section a reference should be given so that the interested reader can easily find more information about the test - what does it look for, what are acceptable parameter values to be used in the test, how does it perform, etc.
**p 5, l 131-133** spike in what variable? From the description it seems to be a spike in $\frac{dT}{dz}$, but that's not clear from the text. Please explain.

**p 5, l 141** possible measurements → possibly correct measurements?

**p 5, l 148** who → which

**p 6, l 179+180** °PSU → PSU

**p 7, l 191** what do you mean by '"hedgehog" type profile'? spiky?

**p 7, l 197** vas → was? or is?

**p 7, l 206** reinforced → increased

**p 8, l 232** If there is a subsection 5.1, why isn't there a subsection 5.2?

**p 8, l 239** lead in → lead to, or: result in

**p 8, l 244** ocean flags → ocean quality flags into account. Apart from this, the sentence is hard to follow and should be reformulated.

**p 8, eq. 1** define $l(i,j)$ and $L(i,j)$

**p 9, eq. 2** define $\overline{T}_p$

**p 9, l 295** differs from → differ by

**p 9, l 301** what is GTSPP? GTS is mentioned before, but what does PP stand for?

**p 9, l 302** erased → deleted

**p 12, l 396** insolation → insulation

**p 12, l 404** contrary to what is stated here, fig. 9 only shows T

**p 13, l 424** as close as possible from the physical measurement → as close as possible to the physical measurement - I am not sure, perhaps better to reformulate the sentence.

**p 13, l 438** something is wrong with the end of this line

**Figures** in all figures the labels are too small. I cannot read them.

**Fig. 2** the colour scale is counter-intuitive. Low values should be blue and high values red.

**Figs. 7+8** why not combine these figures into one?

**Figs. 9+10** why not combine these figures into one?

[Figure]

---

## Referee Comment (RC2) · Anonymous Referee #2 · 28 Jun 2019

**The CORA 5.2 dataset: global in-situ Temperature and Salinity measurements dataset. Data description and validation.**

Tanguy Szekely[1], Jérôme Gourrion[1], Sylvie Pouliquen[2], Gilles Reverdin[3]

[1]Societe Coopérative OceanScope, 115 rue Claude Chape, 29290, Plouzané, Brest

[2]IFREMER, BP 70, Plouzané, 29280, France

[3]Sorbonne -Université, CNRS/IRD/MNHN (LOCEAN), Paris, France. ORCID ID https://orcid.org/0000-0002-5583-8236

Correspondance to: Tanguy Szekely (tanguy.szekely@ocean-scope.com)

**Abstract:** We present the Copernicus in-situ ocean dataset of temperature and salinity (version V5.2). The ocean subsurface sampling varied widely from 1950 to 2017, as a result of changes in the instrument technology and development of in-situ observational networks (in particular, tropical moorings, ARGO program). The global ocean temperature data coverage on an annual basis grows thus from 10% in 1950 (30% for the North Atlantic basin) to 25% in 2000 (60% for the North Atlantic basin) and reaches a plateau exceeding 80% (95% for the North Atlantic Ocean) after the deployment of the ARGO program. The average depth reached by the profiles also increases from 1950 to 2017. The validation framework is presented, and an objective analysis-based method is developed to assess the quality of the dataset validation process. Analyses of the ocean variability are calculated without taking into account the data quality flags (raw dataset OA), with the near real time quality flags (NRT dataset OA) and with the delayed time mode quality flags (CORA dataset OA). The comparison of the objective analysis variability shows that the near real time dataset managed to detect and to flag most of the large measurement errors, reducing the analysis error bar compared to the raw dataset error bar. It also shows that the ocean variability of the delayed time mode validated dataset is almost exempt from the random error induced variability.

**Keywords:** Global dataset, In-situ, Temperature and salinity profiles

[Figure]

[Figure]

**1. Introduction**

Estimating the temperature and salinity ocean state is critical for documenting the evolution of the ocean and its
role in the present climate. To do so, the scientific community relies on in situ measurements at a global scale,
gathered by the scientific community and clustered into global datasets.

[revised manuscript text omitted]
 depth layer and in the 975-1025m depth layer. Most of the temperature profiles flagged in the CORA delayed time mode validation process are indeed XBTs because the XBT instruments are more likely to fail (spikes or bias caused by a stretching of the XBT wire or a contact between the XBT wire and the ship hull), or bad estimation of the measurement depth. These  even if numerous (about 2000 profiles flagged between 2005 and 2016) often have a too small amplitude to induce a large bias between the NRT analysis and the CORA

analysis. Most of the XBTs corrected during this period are moreover T-4 and Deep blue models. These models are usually not measuring in-situ temperature below 460 m depth and 760 m depth respectively. The CORA and NRT variability amplitudes are thus almost the same in the deeper layers.

[revised manuscript text omitted]

---

## Author Comment (AC1) · 8 Aug 2019

In the discussion, the reviewer has emphasized the interest of the paper. He moreover asks to establish a link between the present study and the climatology published by Gouretski et al, 2018 (G2018). G2018 implements an objective analysis method to historical oceanographic measurements to produce a climatology for 1900 to 2015. Most of the measurements G2018 incorprates for the period 1950-2015 are available in the CORA dataset since the CORA dataset is fed by the world ocean database. G2018 differs from the CORA database in that the quality control performed before the objective analysis is fully automated. The comparison of the G2018 and CORA

in term of the number of flagged data and how the differences in flagging impacts the metrics we present in this paper would be very instructive. It would however not been complete without a discussion of the different objectives of the two datasets : providing a most complete set of qualified profiles for scientific studies for CORA whereas G2018 is aimed at producing reliable and robust climatologies and integrated metrics (GOHC for instance). We have chosen not to pursue this comparison as far in order to avoid the pitfall of changing the scope of the present study from describing a new dataset validation method with a metric to estimate the validation process gains to comparing the CORA dataset with others datasets. We have however cited G2018 in the quality control result section to highlight that the amount of flagged profiles is considerably lower using the CORA processing than using G2018 automated methods.

The second point discussed by the reviewer is how we can justify that the CORA dataset in not over flagged. We did not manage to back this conclusion with more evidence. As a result, we have moderated the conclusion of the section Âń CORA quality control results Âż mentioning that we are not able to provide conclusive evidence. There is, indeed, no reference dataset available with only good quality measurements and a global coverage. The comparison of the CORA dataset mean variance variability with the variance variability estimated with other datasets, such as WOD or EN4, is interesting but the difference on the variability level observed then is still insufficient to declare if a dataset is over flagging or not. This particular topic is obviously beyond the scope of this study and will not be developped in the present paper.

Last, the reviewer has pointed out many typos in the original draft. These typos have been corrected and some sentences have been redrafted to ensure a concise and clearer text.

The following points have been raised by the reviewer and corrected in the text.

- p 2, I 5-7 : three times "scientific community" boring, please reformulate
- The sentences have been reformulated
p 2, l 23 : timeseries $\rightarrow$ time scales

- Changed in the text
- p 2, l 25 : Baseline $\rightarrow$ baseline
- Changed in the text

p 4, l 92 : barely $\rightarrow$ slightly? I am not sure what you want to say.

- Changed in the text. The point was to show that the dataset evolution is driven by the improvements in the oceanographic measurement instruments.

p 4, l 96 : barely maintain a plateau at 20% $\rightarrow$ reach a plateau just below 20%? Again lam not sure what you want to say.

- Changed in the text

p 5, I 125 : for each of the test described in this section a reference should be given sothat the interested reader can easily find more information about the test - whatdoes it look for, what are acceptable parameter values to be used in the test, howdoes it perform, etc.

- The test description have been improved.

p 5, I 131-133 : spike in what variable? From the description it seems to be a spike indTdz,but that's not clear from the text. Please explain.

- Done

p 5, l 141 : possible measurements $\rightarrow$ possibly correct measurements?

- Changed in the text

p 5, l 148 : who $\rightarrow$ which

- Changed in the text
p 6, l 179+180 : âŮęPSU→PSU

-Changed in the text

- p 7, I 191 : what do you mean by '"hedgehog" type profile'? spiky?
- Description improved in the text
- p 7, l 197 : vas $\rightarrow$ was? or is?
- Changed in the text
- p 7, l 206 : reinforced $\rightarrow increased$
- Done
- p 8, I 232 : If there is a subsection 5.1, why isn't there a subsection 5.2?
- Removal of the 5.1 subsection title
- p 8, l 239 : lead in $\rightarrow$ lead to, or: result in
- Changed in the text

p 8, l 244 : ocean flags $\rightarrow$ ocean quality flags into account. Apart from this, the sentenceis hard to follow and should be reformulated.

- Changed in the text. A better description of the QC flags is reported on page 5, I 122-125

p 8, eq. 1 : definel(i, j)andL(i, j)

- Done
- p 9, eq. 2 : defineTp
- Done
- p 9, l 295 : differs from $\rightarrow$ differ by
- Changed in the text

p 9, I 301 : what is GTSPP? GTS is mentioned before, but what does PP stand for?

- Updated description
- p 9, l 302 : erased $\rightarrow$ deleted
- Changed in the text
- p 12, I 396 : insolation $\rightarrow$ insulation
- - Changed in the text
- p 12, l 404 : contrary to what is stated here, fig. 9 only shows T
- Changed in the text

p 13, I 424 : as close as possible from the physical measurement  $\rightarrow$  as close as possible to the physical measurement - I am not sure, perhaps better to reformulate thesentence.

- Sentence reformulated
- p 13, I 438 : something is wrong with the end of this line
- Sentence reformulated

Figures : in all figures the labels are too small. I cannot read them.

- Improved figures

Fig. 2the colour scale is counter-intuitive. Low values should be blue and high valuesred.

- The blue and red colors are not related to the low values or high values of the figures. They have not been changed.

Figs. 7+8why not combine these figures into one?Figs. 9+10why not combine these

OSD
**figures into one?**

- The document layout seems to be more clear with splitted figures. The comment have not been taken into account.

Please also note the supplement to this comment: https://www.ocean-sci-discuss.net/os-2018-144/os-2018-144-AC1-supplement.pdf

**OSD**

---

## Author Comment (AC2) · 8 Aug 2019

This review have pointed out many smalls parts of the text that needed to be clarified and/or formulated differently. We managed to do so in the last version of the text, improving the global quality of the text.

Most of these comments concerned the sections describing the dataset and the analysis method. There were however almost no questions relative to the paper discussion and conclusion.

The following points are specifying which action have been take reguarding to the re-

viewer's comments.

p 1, l 3: ARGO only needs a capital A as it is not an acronym – this is true elsewhere in the paper as well.

- Done

p 2, l5 : is should be in

- Done

p 2, l 8 : : 2013 not 3013

- Done

p 2, l 10 : CORA always seems to be in capitals - is it an acronym? If it is, what does it stand for? If it isn't why is it in capitals?

- CORA stands for Coriolis Ocean dataset for ReAnalysis. It has been changed in the text.

p 2, l13 : whereas -> conversely

- Done

p 2, l 21 : Consider putting n and n-1 in italics

- Done

p 2, l 23 : You switch between global heat content, ocean heat content and global ocean heat content quite a lot in this paragraph, it's best to be consistent.

- The text has been accordingly changed.

p 2,l26 : These Cheng and Boyer references don't appear in the reference list.

- The reference have been added in the bibliography section

p 2, l 29 : Inconsistent referencing style.
[Figure]

- Done

p 3, l 50 : Telecommunication

- Done

p 3, l 70 : No need to start a new paragraph here.

- Changed

p 3, 73 : l what about between 1965 and 1970?

- The sentence have been reformulated

p 4, l 91 : Can you explain why the yearly number of XBT profiles strongly decreases?

- Done

p 3, l 110 : Treated as an independent profile as opposed to what? Averaging?

- The sentence have been reformulated

p 6, l160 : Not sure what this sentence is trying to say? What are the redundant tests?

- The sentence have been reformulated

p 6, l 164 : I'm not sure the < signs are the right way round here. I would read this as DEPTH < -2.5m, as in negative depths, but earlier (line 75) you have referenced depths in the ocean as being negative. Please check for consistency.

- The text have been changed for more consistency

p 6 , l 177 : If most of the density inversions are caused by salinity spikes are these also picked up by the spike check or are they too small to be picked up by this?

- The sentence have been reformulated

p 7, l 191 : Are "hedgehog" type profiles very spiky ones? It may be worth clarifying this.

- Some precision have been added.

p 7, l209 : Just the upper and lower adjacent cells in the same grid column or do you look at surrounding grid boxes as well?

- The sentence have been reformulated

p 7, l 212 : Is this not quite a deep continental shelf criterion? Please give a reference explaining this value.

- The sentence have been reformulated

p 7, l 214 : I couldn't find this reference? Is it meant to be in prep for 2019?

- This second paper is under review. The reference have been updated to clarify this point.

p 7, l 205 : Explain what the ISAS objective analysis field does? Makes comparisons to an objective analysis field I assume? Also, spell out ISAS if it is an acronym.

- The exact framework of the ISAS objective analysis method is given in the referenced paper.

p 7, l 224 : Are these different from the stability checks?

- The sentence have been reformulated

p 8, l 228 : Do these tests flag a significant number of data after the minmax test has already been performed?

- The text have been rewriten to answer to this point.

p 8, l 245 : This is the first time you've mentioned different levels of flags. It is probably worth adding in a short paragraph somewhere explaining what flags a profile/ value could get? Is it good, probably good, bad, probably bad?

- The flag level signification have been added in a paragraph in the previous section.

[Figure]

p 8, l 249 : You reference a deeper layer on line 288. If you go down deeper can you specify the depth of these layers as well? The layer depth has been specified.

p 8, l 261 : Are the r_l and r_L always 5 degrees regardless of lat and long values? what are the l(i,j) and L(i,j) the lat and long of the profiles? Although I'm not sure what the i and j stand for - please explain further. Are G(i,j) then the weights given to individual values?

- The correlation function parameters have been explained.

p 8, equation 2 : What's T bar _p? I would have thought the mean temperature of a profile, but this doesn't necessarily make sense in a single grid cell? I'm confused what's being averaged here.

- The sentences have been reformulated to better explain the objective analysis framework.

P 8, l282 : This seems to be repetition of the paragraph on lines 243-245? But with reference to another objective analysis - I suggest merging this paragraph with the earlier one to avoid confusion.

- The paragraphs have been merged.

p 8, l 289 : Do you mean drift in where they're located or drift in the measurement values? Please clarify.

- The signification of drift has been specified.

p 9, l 306 : Do you know why?

- Actually, we do not, but the answer of this question is not within the scope of this study.

p 9, l 351 : Why are these two time periods split when they are contiguous?

- The sentences have been reformulated.

p 11, l 382 : I'm not sure how lines 373-375 link to the rest of the paragraph as it doesn't seem to be explaining the spikes? Having now read on I see that these spikes are explained later, please move that explanation here instead.

- The sentences have been reformulated

p 12, l 385 : Is this over all layers or just the 0-50m layer? Please specify.

- The sentence has been reformulated

p 12, l 387 : which other layers? Those that aren't 75-125m in depth?

- The sentence has been reformulated

p 13, l 434 : You use last twice.

- The sentence has been reformulated

p 13, l 443 : Not entirely sure what these sentences from line 440 mean?

- The sentence has been reformulated

Please also note the supplement to this comment:
https://www.ocean-sci-discuss.net/os-2018-144/os-2018-144-AC2-supplement.pdf

**Supplement:**

[revised manuscript text omitted]

560 **Figure 4: Coriolis database validation process.**

[Figure]

**Figure 5: Coverage of the Temperature and Salinity objective analysis for temperature (dashed line) and salinity (hard line) objective analysis.**

[Figure]

**Figure 6: Percentage of good flags (flags 1 and 2) in the analyzed layers for the NRT dataset (hard line) and for the CORA dataset (dashed line). Upper panel for temperature, lower panel for salinity**

[Figure]

**Figure 7:  Mean salinity standard deviation in the 0-50m layer (top), 75-125 m depth layer (mid.) and 275-325 m depth layer (bot.). The raw dataset (red), NRT dataset (blue) CORA dataset (black) are represented.**

[Figure]

**Figure 8: Mean salinity standard deviation in the 475-525 m depth layer (top), 975-1025 m depth layer (mid.) and 1475-1525 m depth layer (bot.) The raw dataset (red), NRT dataset (blue) CORA dataset (black) are represented.**

[Figure]

575

**Figure 9: Mean temperature standard deviation in the 0-50m layer (top), 75-125 m depth layer (mid.) and 275-325 m depth layer (bot.). The raw dataset (red), NRT dataset (blue) CORA dataset (black) are represented.**

580

[Figure]

**Figure 10: Mean temperature standard deviation in the 475-525 m depth layer (top), 975-1025 m depth layer (mid.) and 1475-1525 m depth layer (bot.) The raw dataset (red), NRT dataset (blue) CORA dataset (black) are represented.**

---

## Author Response (AR1)

**Autor's response to the reviewers**

**First reviewer:**

In the discussion, the reviewer has emphasized the interest of the paper. He moreover asks to establish a link between the present study and the climatology published by Gouretski et al, 2018 (G2018). G2018 implements an objective analysis method to historical oceanographic measurements to produce a climatology for 1900 to 2015. Most of the measurements G2018 incorprates for the period 1950-2015 are available in the CORA dataset since the CORA dataset is fed by the world ocean database. G2018 differs from the CORA database in that the quality control performed before the objective analysis is fully automated. The comparison of the G2018 and CORA in term of the number of flagged data and how the differences in flagging impacts the metrics we present in this paper would be very instructive. It would however not been complete without a discussion of the different objectives of the two datasets : providing a most complete set of qualified profiles for scientific studies for CORA whereas G2018 is aimed at producing reliable and robust climatologies and integrated metrics (GOHC for instance). We have chosen not to pursue this comparison as far in order to avoid the pitfall of changing the scope of the present study from describing a new dataset validation method with a metric to estimate the validation process gains to comparing the CORA dataset with others datasets. We have however cited G2018 in the quality control result section to highlight that the amount of flagged profiles is considerably lower using the CORA processing than using G2018 automated methods.

The second point discussed by the reviewer is how we can justify that the CORA dataset in not over flagged. We did not manage to back this conclusion with more evidence. As a result, we have moderated the conclusion of the section « CORA quality control results » mentioning that we are not able to provide conclusive evidence. There is, indeed, no reference dataset available with only good quality measurements and a global coverage. The comparison of the CORA dataset mean variance variability with the variance variability estimated with other datasets, such as WOD or EN4, is interesting but the difference on the variability level observed then is still insufficient to declare if a dataset is over flagging or not. This particular topic is obviously beyond the scope of this study and will not be developped in the present paper.

Last, the reviewer has pointed out many typos in the original draft. These typos have been corrected and some sentences have been redrafted to ensure a concise and clearer text.

**List of the changes in the text**

- p 2, 1 5-7 : three times "scientific community" boring, please reformulate
  - The sentences have been reformulated
- p 2, 1 23 : timeseries→time scales
  - Changed in the text
- p 2, 1 25 : Baseline→baseline
  - Changed in the text
- p 4, 1 92 : barely $\rightarrow$ slightly? I am not sure what you want to say.
  - Changed in the text. The point was to show that the dataset evolution is driven by the improvements in the oceanographic measurement instruments.

p 4, 196 : barely maintain a plateau at 20%→reach a plateau just below 20%? Again Iam not sure what you want to say.

- Changed in the text

p 5, 1 125 : for each of the test described in this section a reference should be given so that the interested reader can easily find more information about the test - whatdoes it look for, what are acceptable parameter values to be used in the test, howdoes it perform, etc.

- The test description have been improved.

p 5, 1 131-133 : spike in what variable? From the description it seems to be a spike indTdz,but that's not clear from the text. Please explain.

- Done

p 5, 1 141 : possible measurements→possibly correct measurements?

- Changed in the text
- p 5, 1 148 : who→which
  - Changed in the text
- p 6, 1 179+180 : ∘PSU→PSU
  - Changed in the text
- p 7, 1 191 : what do you mean by "hedgehog" type profile'? spiky?
  - Description improved in the text

```
p 7, 1 197 : vas→was? or is?
```

- Changed in the text
- p 7, 1 206 : reinforced→increased
  - Done
- p 8, 1 232 : If there is a subsection 5.1, why isn't there a subsection 5.2?
  - Removal of the 5.1 subsection title
- p 8, 1 239 : lead in→lead to, or: result in
  - Changed in the text

p 8, 1 244 : ocean flags $\rightarrow$ ocean quality flags into account. Apart from this, the sentenceis hard to follow and should be reformulated.

- Changed in the text. A better description of the QC flags is reported on page 5, 1 122-125
- p 8, eq. 1 : definel(i, j)andL(i, j)
  - Done
- p 9, eq. 2 : defineTp
  - Done
- p 9, 1 295 : differs from→differ by
  - Changed in the text
- p 9,1 301 : what is GTSPP? GTS is mentioned before, but what does PP stand for?
  - Updated description
- p 9, 1 302 : erased $\rightarrow$ deleted
  - Changed in the text
- p 12, 1 396 : insolation→insulation
  - Changed in the text
- p 12, 1 404 : contrary to what is stated here, fig. 9 only shows T
  - Changed in the text

p 13, 1424 : as close as possible from the physical measurement  $\rightarrow$  as close as possible to the physical measurement - I am not sure, perhaps better to reformulate thesentence.

- Sentence reformulated
- p 13, 1 438 : something is wrong with the end of this line
  - Sentence reformulated

Figures : in all figures the labels are too small. I cannot read them.

- Improved figures

Fig. 2the colour scale is counter-intuitive. Low values should be blue and high valuesred.

- The blue and red colors are not related to the low values or high values of the figures. They have not been changed.

Figs. 7+8why not combine these figures into one?Figs. 9+10why not combine these figures into one?

- The document layout seems to be more clear with splitted figures. The comment have not been taken into account.

**Second reviewer:**

This review have pointed out many smalls parts of the text that needed to be clarified and/or formulated differently. We managed to do so in the last version of the text, improving the global quality of the text.

Most of these comments concerned the sections describing the dataset and the analysis method. There were however almost no questions relative to the paper discussion and conclusion.

**List of the changes in the text**

p 1, 1 3: ARGO only needs a capital A as it is not an acronym – this is true elsewhere in the paper as well.

- Done
- p 2, 15 : is should be in
  - Done
- p 2, 1 8 : : 2013 not 3013
  - Done

p 2, 1 10 : CORA always seems to be in capitals - is it an acronym? If it is, what does it stand for? If it isn't why is it in capitals?

- CORA stands for Coriolis Ocean dataset for ReAnalysis. It has been changed in the text.
- p 2, 113 : whereas -> conversely
  - Done
- p 2, 1 21 : Consider putting n and n-1 in italics
  - Done

p 2, 1 23 : You switch between global heat content, ocean heat content and global ocean heat content quite a lot in this paragraph, it's best to be consistent.

- The text has been accordingly changed.
- p 2,126 : These Cheng and Boyer references don't appear in the reference list.
  - The reference have been added in the bibliography section
- p 2, 1 29 : Inconsistent referencing style.
  - Done
- p 3, 1 50 : Telecommunication
  - Done
- p 3, 170 : No need to start a new paragraph here.
  - Changed
- p 3, 73 : 1 what about between 1965 and 1970?
  - The sentence have been reformulated
- p 4, 1 91 : Can you explain why the yearly number of XBT profiles strongly decreases?
  - Done
- p 3, 1 110 : Treated as an independent profile as opposed to what? Averaging?

- The sentence have been reformulated

p 6, 1160 : Not sure what this sentence is trying to say? What are the redundant tests?

- The sentence have been reformulated

p 6, l 164 : I'm not sure the < signs are the right way round here. I would read this as DEPTH < -2.5m, as in negative depths, but earlier (line 75) you have referenced depths in the ocean as being negative. Please check for consistency.

- The text have been changed for more consistency

p 6, l 177 : If most of the density inversions are caused by salinity spikes are these also picked up by the spike check or are they too small to be picked up by this?

- The sentence have been reformulated

p 7, 1 191 : Are "hedgehog" type profiles very spiky ones? It may be worth clarifying this.

Some precision have been added.

p 7, 1209 : Just the upper and lower adjacent cells in the same grid column or do you look at surrounding grid boxes as well?

- The sentence have been reformulated

p 7, 1 212 : Is this not quite a deep continental shelf criterion? Please give a reference explaining this value.

- The sentence have been reformulated

p 7, 1 214 : I couldn't find this reference? Is it meant to be in prep for 2019?

This second paper is under review. The reference have been updated to clarify this point.

p 7, 1 205 : Explain what the ISAS objective analysis field does? Makes comparisons to an objective analysis field I assume? Also, spell out ISAS if it is an acronym.

- The exact framework of the ISAS objective analysis method is given in the referenced paper.

p 7, 1 224 : Are these different from the stability checks?

- The sentence have been reformulated

p 8, 1 228 : Do these tests flag a significant number of data after the minmax test has already been performed?

The text have been rewriten to answer to this point.

p 8, 1 245 : This is the first time you've mentioned different levels of flags. It is probably worth adding in a short paragraph somewhere explaining what flags a profile/ value could get? Is it good, probably good, bad, probably bad?

- The flag level signification have been added in a paragraph in the previous section.

p 8, 1 249 : You reference a deeper layer on line 288. If you go down deeper can you specify the depth of these layers as well?

The layer depth has been specified.

p 8, l 261 : Are the r\_l and r\_L always 5 degrees regardless of lat and long values? what are the l(i,j) and L(i,j) the lat and long of the profiles? Although I'm not sure what the i and j stand for - please explain further. Are G(i,j) then the weights given to individual values?

- The correlation function parameters have been explained.

p 8, equation 2 : What's T bar \_p? I would have thought the mean temperature of a profile, but this doesn't necessarily make sense in a single grid cell? I'm confused what's being averaged here.

- The sentences have been reformulated to better explain the objective analysis framework.

P 8, 1282 : This seems to be repetition of the paragraph on lines 243-245? But with reference to another objective analysis - I suggest merging this paragraph with the earlier one to avoid confusion.

- The paragraphs have been merged.
- p 8, 1 289 : Do you mean drift in where they're located or drift in the measurement values? Please clarify.
  - The signification of drift has been specified.
- p 9, 1 306 : Do you know why?
  - Actually, we do not, but the answer of this question is not within the scope of this study.

p 9, 1 351 : Why are these two time periods split when they are contiguous?

- The sentences have been reformulated.

p 11, 1 382 : I'm not sure how lines 373-375 link to the rest of the paragraph as it doesn't seem to be explaining the spikes? Having now read on I see that these spikes are explained later, please move that explanation here instead.

- The sentences have been reformulated
- p 12, 1 385 : Is this over all layers or just the 0-50m layer? Please specify.

- The sentence has been reformulated

p 12, 1 387 : which other layers? Those that aren't 75-125m in depth?

- The sentence has been reformulated
- p 13, 1 434 : You use last twice.
  - The sentence has been reformulated

p 13, 1 443 : Not entirely sure what these sentences from line 440 mean?

- The sentence has been reformulated

[revised manuscript text omitted]